# Single-Cell Transcriptomics in Inherited Retinal Dystrophies: Current Findings and Emerging Perspectives

**DOI:** 10.3390/genes16091088

**Published:** 2025-09-16

**Authors:** Linda Nguyen, Catalina A. Vallejos, Pleasantine Mill, Roly Megaw

**Affiliations:** 1MRC Human Genetics Unit, Institute of Genetics and Cancer, The University of Edinburgh, Crewe Road South, Edinburgh EH4 2XU, UK; catalina.vallejos@ed.ac.uk (C.A.V.); pleasantine.mill@ed.ac.uk (P.M.); 2Princess Alexandra Eye Pavilion, NHS Lothian, Chalmers Street, Edinburgh EH3 9HA, UK

**Keywords:** inherited retinal disease, retinal degeneration, single-cell, transcriptomics, photoreceptor, retinitis pigmentosa, Leber congenital amaurosis, enhanced S-cone syndrome, Stargardt disease, achromatopsia

## Abstract

Inherited retinal dystrophies (IRDs) represent a diverse group of disorders caused by mutations in genes essential for retinal function and maintenance. Traditional bulk RNA sequencing techniques provide valuable information for deciphering disease pathogenesis but lack the resolution to capture variation among specific cell clusters during disease progression. In contrast, single-cell transcriptomics, including single-cell RNA sequencing (scRNA-seq), enables detailed examination of distinct retinal clusters in both healthy and diseased states, uncovering unique gene expression signatures and early molecular changes preceding photoreceptor cell death in IRDs. These insights not only deepen our understanding of the complex pathogenesis of IRDs but also highlight potential targets for novel therapeutic interventions. In this review, we examine the recent literature on the application of single-cell transcriptomics in IRDs to explore how these techniques enhance our understanding of disease mechanisms and contribute to the identification of new therapeutic targets.

## 1. Introduction

Inherited retinal dystrophies (IRDs) represent a clinically and genetically heterogeneous group of disorders characterised by progressive photoreceptor degeneration, ultimately leading to severe vision loss [1]. These conditions affect approximately 1 in 3000 [1,2] to 4000 individuals worldwide [3].

IRDs are characterised by remarkable genetic complexity, with mutations in at least 300 genes identified to date [4], expressed through multiple inheritance patterns including autosomal dominant, autosomal recessive, and X-linked forms [3]. Major IRD subtypes examined in this review include retinitis pigmentosa (RP), Leber congenital amaurosis (LCA), enhanced S-cone syndrome (ESCS), Stargardt disease (STGD), and achromatopsia (ACHM), each presenting distinct clinical phenotypes and underlying genetic causes. Significant phenotypic variability exists even amongst individuals carrying the same genetic mutations [5], creating complex genotype–phenotype relationships that complicate diagnosis and treatment strategies [6].

IRD pathogenesis involves complex interactions between multiple retinal cell types that both respond to and influence the degenerative process [7], with supporting cells such as Müller glia and microglia often exhibiting early transcriptional changes that may precede overt photoreceptor death [7,8]. Characterising cell-type-specific responses is crucial, as disease progression varies highly between distinct retinal cell populations, requiring targeted approaches that can distinguish different cellular responses to disease.

This review reports studies applying single-cell transcriptomics to IRD samples, examining how these approaches have advanced our understanding of disease mechanisms and dynamics with cellular resolution. We present findings by specific IRD subtypes and genetic mutations, highlighting key mechanistic insights and therapeutic targets identified through single-cell approaches. We then identify common themes across different IRD studies and therapeutic implications before identifying current challenges and future considerations for using single-cell technologies in IRD research.

## 2. From Bulk to Single-Cell Transcriptomics in IRDs

RNA sequencing (RNA-seq) has revolutionised the study of gene expression profiles in cells and tissues, enabling comprehensive, high-throughput, and quantitative analysis of the entire transcriptome [9]. The unbiased nature of RNA-seq facilitates the identification of novel genes and pathways without requiring prior hypotheses about specific targets [9,10], making it particularly valuable for identifying novel molecular mechanisms in retinal diseases. RNA-seq provides precise quantification across a wide dynamic range, capturing both abundant and rare transcripts, whilst detecting subtle changes in gene expression [9]. However, traditional bulk RNA-seq approaches have significant limitations when applied to heterogeneous tissues like the retina.

Bulk RNA-seq averages gene expression across all cells in a sample, masking cell-type-specific pathological changes that may be crucial for understanding disease mechanisms [11,12]. Early degenerative events affecting only small subsets of cells become diluted and undetectable in bulk analysis, potentially missing critical early disease signatures that could inform therapeutic interventions [13]. This averaging effect is particularly problematic in retinal tissue, which contains over 100 distinct cell types (with approximately 130 identified in mouse retina [14]), each of which may exhibit distinct responses to disease-causing genetic mutations [11].

Single-cell RNA sequencing (scRNA-seq) overcomes these limitations by capturing gene expression profiles with cellular resolution, providing detailed transcriptome profiling and revealing cellular heterogeneity within complex tissues [11,12]. Computational analyses to cluster cells with similar expression profiles enables the identification of distinct cell populations, facilitating unbiased cell type classification [10], including rare cell types [12] that may be missed in bulk analysis. This single-cell resolution allows investigation of how IRDs affect specific cell types and enables detection of early pathological changes in vulnerable cell populations [12]. Additionally, scRNA-seq can reveal novel cell states, such as stress-responsive or transitional cell populations that emerge during disease progression [11].

The development of single-cell transcriptomics has enabled the creation of comprehensive retinal cell atlases, providing essential reference datasets for understanding both healthy retinal biology and disease-associated changes [11]. Macosko et al. [15] delivered one of the first single-cell transcriptomic atlases of the mouse retina using Drop-seq, establishing a foundational framework for retinal cell type classification. Building upon these foundational studies, Menon et al. [10] created the first human retinal transcriptomic atlas using droplet-based and nanowell platforms, bridging the gap between model organism data and human retinal biology. These atlases have undergone continuous refinement and expansion, now serving as essential reference datasets for interpreting disease-associated transcriptional changes in IRD research.

## 3. Disease-Specific Insights from Single-Cell Transcriptomics

We searched the literature using PubMed to identify relevant studies examining single-cell transcriptomic approaches in IRDs, employing a combination of controlled vocabulary (Medical Subject Headings, MeSH) and free-text terms. The search strategy combined two main groups using the Boolean operator “AND”: single-cell transcriptomic approaches and inherited retinal dystrophies. Single-cell methodologies included “single-cell RNA sequencing”, “scRNA-seq”, “single-nucleus RNA sequencing”, platform-specific terms like “10x Genomics” and “Drop-seq”, and relevant MeSH terms including “Single-Cell Analysis” and “Transcriptome”. IRD terms contained broad categories such as “retinal degeneration” and “inherited retinal dystrophies”, specific conditions including “retinitis pigmentosa”, “Stargardt disease”, and “Leber congenital amaurosis”, plus corresponding MeSH terms such as “Retinal Degeneration”, “Retinal Dystrophies”, and “Eye Diseases, Hereditary”. We restricted the search terms to title and abstract fields ([tiab]) and limited the results to English-language publications with full-text availability, published between 2019 and 2025 to capture recent technological advances in single-cell transcriptomics.

The initial PubMed search yielded 412 results on 20 June 2025. Studies were included if they utilised single-cell transcriptomics methodologies to investigate IRDs. We excluded studies examining exclusively healthy retinal samples to maintain focus on pathological processes in IRDs. After screening, 27 papers met the inclusion criteria, representing 23 distinct datasets (some papers shared data from the same experiment). These 23 studies form the basis for this review and are summarised in Table 1. The following sections present gene-specific findings grouped by disease type: RP (with subsections on *RHO*, *PDE6B*, *RPGR*, *USH*, and other RP gene mutations), LCA, ESCS, STGD, and ACHM (Figure 1).

### 3.1. Retinitis Pigmentosa

Retinitis pigmentosa (RP) is the most common IRD [2,43], affecting more than 1.5 million people worldwide [43] with a prevalence of approximately 1 in 4000 individuals [3,5,43]. RP is the leading cause of inherited blindness in working-age adults and has substantial impacts on life quality of patients and healthcare systems [3,43].

The progression of RP varies between individuals but classically begins with night blindness, followed by progressively narrowing visual fields (tunnel vision) and can ultimately lead to the loss of central vision [3]. While some individuals experience noticeable vision loss during childhood, others remain free of symptoms until reaching mid-adulthood, making age of onset an unreliable indicator of disease severity [3,5,6]. In most typical RP cases, the decline in rod photoreceptors exceeds that of cone photoreceptors, though some instances show cone–rod degeneration patterns with reduced visual acuity and impaired colour vision [3].

RP is genetically heterogeneous, with the majority of cases being monogenic [3] and linked to mutations in more than 80 genes [5]. In most cases, RP remains restricted to the eye (non-syndromic RP) [5], whilst 20–30% of RP patients present with an associated non-ocular condition (syndromic RP) [5]. The most common syndromic forms include Usher syndrome (USH; characterised by hearing impairment) [3,5] and Bardet–Biedl syndrome (involving obesity, renal abnormalities, and cognitive impairment) [3,5].

Currently, no curative treatments are available that can stop RP progression or restore vision [43]. Given that RP-associated genes have different localisations and functions across retinal cell types [3,5], there is a critical need for molecular-level understanding to develop personalised therapeutic approaches targeting specific cellular pathways. The following sections present findings from single-cell studies organised by the specific RP-associated genes for which we found transcriptomic studies.

#### 3.1.1. *RHO* Mutations

Mutations in the rhodopsin gene (*RHO*), first identified in 1990 by Dryja et al. [5,44,45], are associated with autosomal dominant RP (adRP) [5], accounting for approximately 25% of adRP cases [3,6,46]. The clinical course is highly variable, with onset typically before age 10 and legal blindness often developing between 50 and 80 years [5]. Additionally, mutations in *RHO* are also associated with congenital stationary night blindness (CSNB), a nonprogressive condition [5].

*RHO* encodes the G protein-coupled receptor rhodopsin, which is activated by photons and initiates the phototransduction cascade in the outer segment of rod photoreceptors [45]. Over 200 mutations in *RHO* have been identified, with the P23H mutation being the most common mutation in the human *RHO* gene [45] and R135W representing another missense variant affecting rhodopsin structure [47]. P23H results in pronounced misfolding and endoplasmic reticulum (ER) retention [45,48], whereas R135W causes partial misfolding, disrupting interactions with signalling proteins [47]. Both mutations trigger ER stress, elevate levels of reactive oxygen species, and ultimately cause photoreceptor cell death through rhodopsin accumulation in the ER [45,48,49].

Single-cell transcriptomics studies have examined these molecular defects across disease progression stages using complementary model systems: P23H mouse models during active degeneration (postnatal weeks 7 and 10) [16,17], adult zebrafish to understand cell-type-specific changes [18], and human iPSC-derived retinal organoids (rOrgs) at immature (D120) and mature (D270) stages [19].

During active degeneration, shrinking numbers of mature rod photoreceptors [16,18,19] showed functional decline through decreased expression of phototransduction genes and disrupted structural integrity through reduced expression of genes involved in the photoreceptor cilium, inner/outer segments, and overall photoreceptor development [16]. Elevated stress responses included upregulated misfolded protein responses, oxidative stress markers, and metabolic shifts toward enhanced glycolysis [18].

In response to oxidative stress, zebrafish cones, retinal pigment epithelium (RPE) cells, and bipolar cells showed upregulated antioxidant genes [18], whilst mouse Müller glia exhibited compensatory responses by upregulating photoreceptor maintenance genes and, during advanced degeneration, co-expression of rod/cone markers [16]. Patient organoids demonstrated enhanced cell–cell communication networks and increased astrocyte proportions with activated inflammatory signalling [19]. Therapeutic intervention with fibroblast growth factor 21 (FGF21) in *RHO* mice activated axon development and synaptic remodelling pathways in Müller glia, suggesting improved glial function through the serum response factor (SRF) pathway. SRF motifs were enriched in FGF21-responsive genes, with *Srf* specifically expressed in Müller glia and astrocytes and increased by FGF21 treatment [17].

Unlike mammalian models, the zebrafish P23H model uniquely demonstrated ongoing regenerative capacity alongside degeneration, with dramatically increased retinal progenitor cells and newly forming rods, indicative of continued regeneration [18]. Müller cells shifted transcriptionally towards a proliferative state, whilst microglial populations showed distinct roles in phagocytic clearance and apoptosis regulation to facilitate regeneration [18].

#### 3.1.2. *PDE6B* Mutations

Rod phosphodiesterase 6 (PDE6) consists of two catalytic subunits (PDE6A and PDE6B) and two inhibitory gamma subunits (PDE6G) [50], with both catalytic subunits essential for full enzymatic activity [51]. PDE6 plays an important role in rod phototransduction by hydrolysing cGMP following light activation of rhodopsin and transducin, leading to closure of cGMP-gated ion channels and membrane hyperpolarisation [50]. *PDE6B* mutations lead to PDE dysfunction, resulting in failure to hydrolyse cGMP and subsequent cGMP accumulation in the retina [3].

*PDE6B* mutations cause autosomal recessive RP (arRP) with rod–cone dystrophy patterns, accounting for roughly 2–4% of arRP cases, and autosomal dominant CSNB [5,52,53,54]. Clinical onset typically occurs before age 10 [5,54], with significant peripheral visual field loss manifesting between 10 and 30 years [5].

Single-cell studies have extensively examined *rd1* and *rd10* mouse strains carrying nonsense and missense *Pde6b* mutations, respectively, with the *rd1* strain showing earlier onset (P7–10) [55] and faster progression (complete photoreceptor loss by P17–21) [56,57] compared to *rd10* mice (onset P16–18 [58,59] and complete loss P40–45 [59,60]).

Rod photoreceptors showed severe degeneration (55% to 16% population reduction), with consistently downregulated *Pde6b* and dysregulated phototransduction, apoptosis, and calcium signalling pathways [20,21,23,24,25].

Abnormal calcium (Ca^2+^) signalling represented a core pathological mechanism, with rods showing more disruption than cones [20,23]. These changes appeared early before cell death [25]. From early to peak degeneration, upregulated cyclic nucleotide-gated (CNG) channel genes [20,23] and voltage-gated calcium channel (VGCC) genes [23] drive heightened Ca^2+^ influx. Increased sodium–calcium exchanger (NCX) genes (e.g., *Slc8a1*) [20,23] and downregulated calcium release-activated channels (CRACs) [23] suggest compensatory responses to elevated Ca^2+^ levels. Late phases showed reduced CNG channel expression but an upregulated protein kinase G (PKG) pathway (*Prkg1*) [20], with T-type VGCC and NCX pathways validated as therapeutic targets in explant cultures [23]. Overall, modulating Ca^2+^ channels or inhibiting PKG may preserve photoreceptor viability.

Ca^2+^-dependent mitogen-activated protein kinase (MAPK) pathways (ERKs, JNKs, and p38 kinases) showed upregulation in *rd1* rods, with MAPK/c-Jun and JNK signalling validated as therapeutic targets [20,21,24,61,62]. Downstream, early growth response 1 (EGR1) functions as a critical transcriptional mediator of cellular stress responses [63], with consistent but temporally distinct patterns: sustained upregulation in *rd1* rods [21] versus transient early expression in *rd10* rods [25]. *Egr1* regulated early degenerative genes as both an activator and repressor depending on degeneration stage, whilst *Cebpd* became highly upregulated in late phases as a potential neuroprotective mechanism [25].

Integration of single-cell transcriptomics with epigenetic analysis identified chromatin-modifying enzymes histone deacetylase (HDAC) and poly (ADP-ribose) polymerase (PARP) as regulators of non-apoptotic cell death [20,22], with cell-type-specific expression patterns: high *Hdac1* in late-stage *rd1* rods and reduced *Parp1*, particularly in cones [20,22]. Conversely, Dong et al. [22] demonstrated that *Pde6b* mutations triggered HDAC overactivation, causing chromatin condensation and PARP overactivation, with suberoylanilide hydroxamic acid (SAHA) treatment delaying photoreceptor loss through dual-pathway inhibition [22].

During peak [25] and late degeneration [20] in both *rd1* and *rd10* rods, cones showed early transcriptomic responses preceding secondary death, with *rd10* cones more similar to degenerating rods than wild-type cones. Cone responses included ER stress, unfolded protein response, mitochondrial dysfunction, and increased tubulin cytoskeleton transcripts [20,25].

Beyond the primary photoreceptor dysfunction and secondary cone degeneration, single-cell transcriptomics revealed extensive tissue-level responses involving glial and immune populations. Müller glia showed strong gliotic responses with upregulated *Gfap*, increased transcripts, and higher mitochondrial/ribosomal content [26], as well as greater proliferative capacity alongside rods [21]. Upregulated major histocompatibility complex (MHC) components and complement system genes suggest antigen-presenting and innate immunity functions [26]. Microglial activation was marked by elevated *Cd68* expression and dramatic population expansion from 0.34% to 3.94% in diseased retinas [24].

Metabolic profiling revealed complex disruptions across retinal cell types in both *rd1* and *rd10* mice. Cones consistently showed downregulated glycolysis genes [20,25], whilst rod responses varied by model: downregulated glycolytic enzymes in early *rd10* phases [25] versus upregulated glycolysis and oxidative phosphorylation (OXPHOS) genes in *rd1* rods [20]. Both models eventually showed mitochondrial dysfunction with downregulated respiratory complex genes [20,25]. Müller cells demonstrated enhanced OXPHOS and metal homeostasis genes, suggesting compensatory metabolic support and antioxidant defence [20,26].

In summary, *PDE6B*-mediated RP involves elevated cGMP levels causing abnormal Ca^2+^ influx, which activates MAPK/JNK/EGR1 and HDAC/PARP pathways, triggering metabolic reprogramming and coordinated cell death responses. Primary rod death and secondary cone degeneration occur alongside compensatory Müller glial responses and broader immune activation.

#### 3.1.3. *RPGR* Mutations

Mutations in the retinitis pigmentosa GTPase regulator (*RPGR*) gene are responsible for X-linked retinitis pigmentosa (XLRP), accounting for approximately 70–90% of XLRP cases and 10–20% of all RP cases [3,64,65,66]. RPGR interacts with actin-binding proteins to regulate outer segment maintenance and protein trafficking in the connecting cilium, essential for facilitating rhodopsin transport to the outer segments [67,68]. As RPGR is located on the X chromosome, males are more severely affected, whilst female carriers can exhibit variable symptoms ranging from mild to severe depending on patterns of X-chromosome inactivation [69]. *RPGR* mutations are associated with early onset of central vision loss before age 5 and a mean age of legal blindness of 45 years [5,70]. Known syndromic associations include hearing loss and respiratory infections [5,71,72]. Both rods and cones can be affected and several IRD phenotypes are linked to the disease, including cone–rod dystrophy (CRD), cone dystrophy (CD), and macular dystrophy (MD) [5,73,74,75,76]. The *RPGR* gene encodes 10 different transcripts through alternative splicing, 5 of which are protein-coding. Two major isoforms are especially important: RPGR^Ex1-19^ and RPGR^ORF15^. The constitutive RPGR^Ex1-19^ isoform spans 19 exons and is ubiquitously expressed, whilst the RPGR^ORF15^ isoform contains a unique ORF15 exon with highly repetitive sequence coding for glutamate–glycine repeats [66]. Although RPGR^ORF15^ has a restricted expression pattern that includes the retina, where it is predominantly expressed, recent evidence demonstrates its importance in other ciliated tissues, including airway epithelia where it regulates motile cilia function [77]. Despite its widespread expression, RPGR^Ex1-19^’s unique exons have not been associated with human disease. In contrast, RPGR^ORF15^ is critical for both photoreceptor function and survival as well as motile cilia regulation, with up to 60% of *RPGR* mutations occurring within the ORF15 region [66,77].

Despite the high frequency of *RPGR* mutations, only two single-cell studies were identified. Li et al. [28] established patient-derived iPSC organoid models, analysing developmental stages D40-D200 and documenting cell composition differences between control and *RPGR* patient organoids. Newton et al. [27] provided deeper mechanistic insights by combining single-cell transcriptomics with multiple complementary validation methods using *RPGR*-mutant mouse models (*Rpgr^Ex3d8^* and *Rpgr^ORFd5^*).

Pseudotime analysis mapped disease progression from healthy to degenerating rod photoreceptors, revealing progressive downregulation of phototransduction genes and upregulated PI3K/AKT signalling. Dysregulated autophagy showed increased autophagy-related genes and lysosome biogenesis, leading to mitochondrial dysfunction with upregulated mitochondrial genes as compensatory responses. Single-cell analysis revealed highly upregulated necroptosis genes, identifying necroptosis as the predominant cell death mechanism and therapeutic target.

In summary, the mouse study from Newton et al. [27] provides detailed mechanistic insights into *RPGR* pathogenesis, revealing autophagy disruption, mitochondrial stress, and necroptotic cell death, whilst the organoid study from Li et al. [28] establishes a valuable human disease model system and demonstrates developmental and transcriptomic changes in *RPGR*-mutant photoreceptors.

#### 3.1.4. *USH* Mutations

Usher syndrome (USH) represents the most frequent syndromic form of RP, accounting for 20–40% of arRP cases [3]. It is associated with hearing impairment and is classified into three clinical subtypes based on severity, age of onset, and vestibular involvement, each of which is linked to a specific genetic cause [78,79]. Among these subtypes, USH1 represents the most severe form, with congenital hearing loss, loss of vestibular function, and early-onset RP typically manifesting before puberty [79,80,81]. USH2 is the most common subtype, representing over half of all USH cases, and is characterised by moderate-to-severe congenital hearing impairment, intact vestibular function, and post-pubertal onset of RP [79,80,81]. USH3 represents the least common USH subtype, characterised by progressive hearing loss appearing from late childhood to age 35, varying degrees of vestibular impairment, and the onset of RP usually by the second decade of life [81]. USH proteins play important roles in maintaining the structure and function of sensory cells by building protein networks (molecular scaffolds) that coordinate cellular transport and signalling. They help maintain normal hearing, balance, and vision [82].

In our literature screen, we found only one study by Leong et al. [29] that definitely meets all inclusion criteria (Table 1). The authors performed single-cell transcriptomics on retinal organoids derived from induced pluripotent stem cells (iPSCs) from Usher syndrome type 1B (USH1B) patients and controls, representing the first human RP model for USH1B in a pre-symptomatic disease state consistent with childhood onset. The disease is caused by a mutation in the myosin VIIA (*MYO7A*) gene, which encodes an ATP-dependent myosin motor protein that travels along actin filaments, transporting and anchoring protein complexes [83,84,85].

While *MYO7A* has traditionally been associated with RPE function, the scRNA-seq analysis by Leong et al. [29] detected *MYO7A* expression in Müller cells, bipolar cells, rods, and cones. USH1B rods showed GO enrichment for chemical and oxidative stress responses and upregulated pro-apoptotic and antioxidant markers, whilst cones showed no significant molecular pathology. These rod-specific stress responses aligned with clinical observations of primary rod degeneration in USH1B-RP. Müller cells showed differentially expressed genes enriched for apoptotic signalling regulation, suggesting potential vulnerability of this cell type.

#### 3.1.5. Other RP Gene Mutations

Beyond the better-characterised gene mutations in *RHO*, *PDE6B*, *RPGR*, and *USH*, studies on other RP-associated gene mutations remain relatively sparse. Here, we focus on studies for five RP genes, *CWC27*, *PRPF8*, *CRB1*, *PROM1*, and *ADIPOR1*, for each of which we only identified one relevant paper (Table 1). Despite the limited number of studies, these genes represent a diverse spectrum of molecular functions and pathophysiological mechanisms. Together, they illustrate how defects in distinct processes (i.e., pre-mRNA splicing, retinal cell polarity, photoreceptor disc morphogenesis, and metabolic homeostasis) can each culminate in an RP-like phenotype.

Mutations in the spliceosomal gene *CWC27* can cause non-syndromic and syndromic forms of arRP. Clinical phenotypes include retinal degeneration, skeletal defects, and neurological defects [86,87]. CWC27 encodes a spliceosome-associated protein crucial for pre-mRNA splicing [88].

Bertrand et al. [30] provided the first single-cell transcriptomics analysis of *Cwc27*-associated retinal degeneration using a mouse model at early disease stages. The analysis provided cellular-level insights into how CWC27 splicing defects primarily impact rod photoreceptor mitochondrial function and trigger Müller glial inflammatory responses, while other retinal cell populations showed minimal changes.

Another splicing factor mutation associated with retinal degeneration involves PRPF8. *PRPF8* mutations cause adRP type 13 (RP13), accounting for approximately 11% of adRP cases and representing one of the most frequent causes of IRD [31,46], leading to early-onset RP typically before age 10 [5]. *PRPF8* encodes the highly conserved scaffolding protein PRPF8, a precursor mRNA processing factor (PRPF) that is part of the spliceosome [46,89,90,91]. Despite its ubiquitous expression and essential role in pre-mRNA splicing, mutations in *PRPF8* lead to retina-specific pathology [31,90].

Atkinson et al. [31] provided the first comprehensive single-cell analysis using RP13 patient-derived iPSCs, revealing higher proportions of degenerating rods in retinal organoids. Both photoreceptor types showed significant downregulation of cell-type markers and phototransduction genes, with pathway analysis identifying phototransduction and photoreceptor degeneration as the top disrupted processes. These findings complement broader tissue comparison studies (i.e., bulk RNA-seq across iPSCs, kidney organoids, RPE, and retinal organoids), providing cellular-level resolution of how PRPF8 splicing dysfunction translates into photoreceptor-specific pathology.

Beyond splicing-related mechanisms, cellular polarity can impact photoreceptor viability, as shown by *CRB1*. *CRB1* encodes a large transmembrane protein that functions as a regulator of apical–basal cell polarity and cell adhesion between photoreceptors and Müller cells in the retina [92,93,94]. Mutations in the Crumbs homolog 1 (*CRB1*) gene can cause the early-onset IRDs arRP and LCA [5,94]. Loss of CRB1 function disrupts critical cellular junctions and leads to photoreceptor degeneration [93,94,95].

Boon et al. [32] employed scRNA-seq on *CRB1* patient-derived iPSC retinal organoids, identifying endosomal dysfunction as the primary pathological mechanism in rods and Müller cells, alongside rod phototransduction defects. Müller cells showed pathway dysregulation, including the endosomal system, cell maintenance/motility/adhesion, protein binding, cell death, and iron transport/phosphorylation. AAV-mediated gene therapy partially restored endosomal gene expression patterns and increased photoreceptor numbers, leading to identification of endosomal recycling proteins as potential therapeutic targets.

Another transmembrane protein implicated in RP is PROM1 (prominin-1/CD133), which plays a crucial role in photoreceptor outer segment disc morphogenesis [96,97]. Mutations in *PROM1* disrupt the membrane organisation and shedding of outer segment discs, impairing the continuous renewal of photoreceptor cells. This structural dysfunction compromises photoreceptor function, ultimately resulting in vision loss [96,97]. They are associated with arRP [98,99] but can also cause autosomal dominant MD [5,100].

Shigesada et al. [33] used scRNA-seq in pre-symptomatic *Prom1*-deficient mice, capturing immediate transcriptional responses before morphological changes. Visual function genes were downregulated in photoreceptors, whilst stress responses involved cell–cell communication through upregulated *Edn2* in rod signalling to glial cells, and widespread stress markers across multiple cell types were observed. Rod-specific *Igf1* downregulation led to the identification of two therapeutic strategies, metabolic support via IGF1 and anti-inflammation via endothelin blockade, both validated through AAV-IGF1 gene therapy and endothelin antagonist therapy (bosentan) improving retinal function.

The final RP gene for which we found a single-cell transcriptomics study is *ADIPOR1*, linking retinal degeneration to disrupted metabolic homeostasis [101]. *ADIPOR1* encodes the adiponectin receptor protein 1 and is involved in cell signalling pathways that regulate energy homeostasis, inflammation, and cell survival [102]. Though widely expressed, ADIPOR1 shows the highest protein levels in neural tissues, particularly photoreceptors and RPE cells [103]. ADIPOR1 possesses intrinsic ceramidase activity [102,104] that breaks down toxic ceramides [105,106]. Mutations have been associated with both non-syndromic and syndromic forms of RP [107,108], as well as being identified as a genetic risk factor for advanced age-related macular degeneration [109].

Lewandowski et al. [34] conducted comprehensive single-cell analysis of *Adipor1* knockout mice across multiple disease stages, revealing ADIPOR1 as the most abundant retinal ceramidase. Its absence caused ceramide accumulation and progressive photoreceptor death, identifying ceramide metabolism as a therapeutic target. This led to successful pharmacological intervention with FDA-approved drugs (desipramine/L-cycloserine) that reduced ceramide levels and provided neuroprotection.

### 3.2. Leber Congenital Amaurosis

Among the IRDs, Leber congenital amaurosis (LCA) represents one of the most severe forms [110], characterised by profound visual impairment that typically presents at or in early infancy [111]. It accounts for approximately 5% of all IRDs and 20% of childhood blindness worldwide [111].

The clinical presentation can vary but typically includes severe visual impairment, reduced or absent electroretinography (ERG) responses, involuntary eye movements (nystagmus), compromised or absent pupillary responses, and eye pressing or poking with fingers (oculodigital sign) [111]. The exact clinical phenotype varies depending on the specific gene affected [111].

Most of the LCA genes follow an autosomal recessive inheritance pattern [110], but some cause autosomal dominant forms (such as *CRX*) [110,111]. There are 19 distinct LCA subtypes (LCA1-LCA19) caused by genetic variants in different genes, with over 25 genes associated with the condition [112,113]. Among these, three genes have been studied using single-cell transcriptomics approaches: *CEP290, RPE65,* and *CRX*.

CEP290 (290 kDa centrosomal protein) represents one of the most frequently mutated genes in LCA [114], accounting for 15–30% of all LCA cases and designated as LCA10 [110,113,115,116]. It encodes a centrosomal protein that localises to the connecting cilium of photoreceptors, where it plays an important role in protein trafficking between inner and outer segments [110,117]. Mutations in *CEP290* disrupt ciliary function and protein transport, leading to photoreceptor degeneration [117].

Fogerty et al. [35] used scRNA-seq in zebrafish *cep290* mutants, revealing that unlike typical zebrafish regenerative responses [118], Müller glia remained in rest-associated states despite active immune responses during progressive degeneration. Although regeneration-promoting genes *stat3* and *yap1* were upregulated, persistently high *notch3* expression blocked Müller glia reprogramming, as Notch signalling must be downregulated for successful regeneration [119,120]. The study identified a novel Müller glia state with simultaneous quiescence and reactivity markers, suggesting that sustained Notch signalling prevents regeneration despite inflammatory signals and highlighting *notch3* as a potential therapeutic target for *CEP290*-LCA10.

*RPE65*-associated LCA (LCA2) accounts for 4–16% of LCA cases [110,115,121,122,123] and 2% of arRP [121,124]. *RPE65* encodes the retinal pigment epithelium-specific 65 kDa protein, a retinoid isomerase that is critical in the visual (retinoid) cycle [125]. Loss-of-function mutations in *RPE65* disrupt the visual cycle, leading to childhood-onset retinal degeneration [126,127]. RPE65-associated LCA2 became the first IRD to receive approved gene therapy [128], but long-term studies have revealed continued retinal degeneration despite initial visual improvements [129]. Recent advances in base editing technology have enabled direct correction of these mutations in mouse models [36,130], offering a potentially more durable therapeutic approach than conventional gene augmentation strategies.

Choi et al. [36] demonstrated successful *in vivo* base editing correction of *RPE65* mutations in LCA2 mice (*rd12*), with treated cones showing gene expression profiles more similar to wild-type. Single-cell analysis revealed restoration of key cone phototransduction genes and upregulated neuroprotective genes whilst showing downregulated cell death markers. Long-term follow-up at 6 months demonstrated sustained cone function and survival, indicating durable therapeutic correction and highlighting base editing’s potential for treating LCA2 and other IRDs.

Mutations in the cone–rod homeobox gene (*CRX*) represent a notable exception to the predominantly recessive forms of LCA discussed above. *CRX* mutations account for 0.6–2.8% of LCA [115,122,123] but can also cause CRD and MD [37]. Although *CRX* mutations primarily lead to autosomal dominant LCA7 [115,131,132], both dominant and recessive inheritance patterns have been reported [111,133]. *CRX* encodes a photoreceptor-specific transcription factor that is essential for both rod and cone photoreceptor development and function [116,131]. Mutations lead to early-onset severe visual impairment with functional deficits seen in ERGs [134].

Kruczek et al. [37] used patient-derived retinal organoids with scRNA-seq to study *CRX*-LCA7 and evaluate AAV-mediated gene augmentation therapy. Single-cell analysis revealed clear separation between patient and control populations, with AAV-treated cells exhibiting an intermediate gene expression pattern. Key photoreceptor genes were partially rescued upon treatment, with therapeutic effects persisting for 6 months without toxicity, demonstrating gene augmentation efficacy at the single-cell level.

Apart from the studies presented here, the study from Boon et al. [32] also investigated the *CRB1* mutation that causes both arRP and LCA and was therefore presented earlier in the RP section. This study similarly combined scRNA-seq with AAV-mediated gene therapy, demonstrating the broader applicability of single-cell approaches for evaluating therapeutic interventions across multiple LCA subtypes. The integration of single-cell data with therapeutic interventions has proven particularly valuable in LCA research, enabling molecular assessment of treatment efficacy and identification of biomarkers for therapeutic response.

### 3.3. Enhanced S-Cone Syndrome

Enhanced S-cone syndrome (ESCS) is a rare IRD, characterised by an overabundance of the short-wavelength (S) cones alongside rod dysfunction [135,136,137].

It follows an autosomal recessive inheritance pattern [138,139], with typical symptoms including night blindness (nyctalopia) at an average age of 4 years, light sensitivity (photophobia), and colour vision anomalies, with increased sensitivity to blue light [139]. Patients’ ERGs have a characteristic pattern with reduced rod and cone responses but a strong S-cone response [136,138].

Known causative genes include *NRL* (neural retina leucine zipper) or *CRX* [140], but the most frequently implicated gene is *NR2E3* (nuclear receptor subfamily 2, group E, member 3), which accounts for around 94% of ESCS cases [135]. The encoded NR2E3 protein is a photoreceptor-specific transcription factor that plays a crucial role in retinal development [140] by suppressing cone differentiation and promoting rod photoreceptor specification [141,142]. NR2E3 dysfunction causes rod precursors to develop into S-cone-like photoreceptors and become the predominant cone subtype [143].

Two studies published in 2024 have applied single-cell transcriptomics to better understand *NR2E3*-related pathogenesis.

Mullin et al. [38] used scRNA-seq on *NR2E3* patient organoids across developmental stages (D40-D260), identifying D80 as the photoreceptor commitment point. They discovered novel “divergent rods” emerging at D120, unique to *NR2E3*-null organoids, which abnormally co-expressed rod and cone genes but lacked critical *RHO* expression despite normal *NRL*. This demonstrated that *NR2E3* loss affects rod maturation rather than commitment. AAV gene therapy administered after divergent rod formation (D130-D160) showed only a 22% response rate, highlighting the need for early intervention before divergent rods emerge.

Complementing the human organoid study, Aísa-Marín et al. [39] employed two different *Nr2e3* mutant mouse models, similarly identifying “hybrid” photoreceptors co-expressing rod and cone markers, particularly increased S-opsin, with decreased differentiated cones (70% to 20%) and increased hybrid cones (20% to 40–50%). Like Mullin et al. [38], they found altered differentiation pathways rather than complete fate conversion, as well as retinal remodelling responses and non-apoptotic cell death mechanisms including necroptosis and parthanatos. One major difference between both studies was retained rhodopsin expression in mouse rods (unlike complete absence in human organoids), likely reflecting hypomorphic rather than complete null mutations with residual NR2E3 protein function.

Together, these scRNA-seq studies have redefined ESCS as a disorder of photoreceptor maturation rather than commitment, revealing hybrid cell states and critical therapeutic windows to guide future gene therapy strategies.

### 3.4. Stargardt Disease

Stargardt disease (STGD) represents one of the most common inherited juvenile-onset MDs, with 5.5 million individuals affected worldwide [144,145].

The condition is inherited in an autosomal recessive manner, typically beginning during childhood [144,145], and is characterised by progressive central vision loss due to lipofuscin accumulation in the RPE and subsequent photoreceptor degeneration [146,147]. Other characteristics include deficiency in colour vision, central macular atrophy, and yellow-white flecks in the RPE [145].

More than 95% of STGD cases are caused by mutations in *ABCA4* (ATP-binding cassette subfamily A member 4; STGD1) [148], which encodes a protein localised to the disc membranes of photoreceptors. This protein removes toxic retinal derivatives from disc membranes, enabling their clearance from the cell. When *ABCA4* is defective, toxic compounds remain and react with membrane lipids, leading to lipofuscin accumulation in the RPE and subsequent progressive retinal degeneration [149,150].

Two 2025 studies used scRNA-seq in *ABCA4* patient iPSC organoids to model STGD1, both finding photoreceptor structural alterations. Zhao et al. [41] conducted a longitudinal analysis across five developmental time points (D40-D260), showing cellular evolution from progenitors to rod/cone predominance and reduced cone populations with morphological abnormalities. Watson et al. [40] focused on late-stage organoids (D200), discovering spatial mislocalisation of photoreceptor cells when performing IHC, appearing in both basal and apical positions. Single-cell data showed phototransduction disruption, impaired maturation, and upregulated stress pathways but minimal apoptosis, suggesting that cellular stress rather than death drives mislocalisation. Both studies established valuable platforms for drug development and future STGD research.

### 3.5. Achromatopsia

Achromatopsia (ACHM) is a rare autosomal recessive IRD, in which cone dysfunction leads to severely reduced or absent colour vision, photophobia, nystagmus, and severely reduced visual acuity [151,152]. In the majority of ACHM cases, cones are present in the retina but lack the ability to respond to light [153]. It typically presents at birth or early infancy and affects approximately 1 in 30,000–50,000 individuals worldwide [151,152,154].

Most ACHM cases (up to 70–90%) are due to mutations in *CNGA3* or *CNGB3*, whilst fewer than 10% of cases are caused by mutations in *GNAT2*, *PDE6C*, *PDE6H*, and *ATF6* [151,155,156,157]. Although *PDE6C* represents one of the less common genetic causes of ACHM (1.4–2.4% [154,158]), its study provides crucial insights into cone phototransduction mechanisms. The gene encodes the catalytic alpha subunit of cone-specific phosphodiesterase, an enzyme responsible for hydrolysing cGMP to terminate the phototransduction signal and restore the cone to its dark-adapted state [158]. Mutations in *PDE6C* disrupt this critical step in the phototransduction cascade, resulting in loss or dysfunction of cones and the typical ACHM phenotype [158]. Given that PDE6C operates within the same functional pathway as CNGA3, CNGB3, GNAT2, and PDE6H [152], therapeutic interventions targeting this mechanism may offer benefits across multiple genetic forms of ACHM.

Our literature search revealed only one study that examined diseased ACHM samples using single-cell transcriptomics. Miller et al. [42] investigated HDAC inhibition effects on cone degeneration in ACHM, discovering that diseased cones had decreased H3K27me3 (trimethylation of histone H3 protein at lysine 27 residue) marks. They tested histone H3 lysine 27 (H3K27) demethylase inhibition in *Pde6c*-mutant mice, with scRNA-seq revealing beneficial transcriptional changes, including suppressed oxidative phosphorylation, mitochondrial dysfunction, and ER stress pathways. Despite improved transcriptional changes, single injection showed no histological improvement, whilst continuous treatment in retinal explants increased cone survival and improved opsin localisation, indicating that sustained H3K27 demethylase inhibition is required for cone protection.

## 4. Common Themes and Therapeutic Implications

We organised the studies by disease type and genetic mutation for presentation (Figure 1), recognising that certain mutations may be associated with overlapping disease phenotypes. Our literature search identified 15 studies investigating mutations associated with RP, comprising 4 studies of *RHO* mutations, 3 of *PDE6B* mutations, 2 of *RPGR* mutations, and 6 examining other RP-related mutations. Additionally, three studies investigated LCA-associated mutations, whilst two studies each examined ESCS- and STGD-associated mutations, and one study focused on an ACHM-associated mutation. Most studies were performed in the USA (9 out of 23), followed by China (5 studies) and the UK (4 studies), whilst the Netherlands, Australia, Japan, Spain, and Switzerland each contributed 1 study. In terms of model systems, 12 studies used mouse models, 2 used zebrafish, and 9 studied human iPSC-derived retinal organoids. The majority employed 10x Genomics technology for their studies (19 out of 23), consistent with its robust performance profile amongst high-throughput single-cell platforms [159].

Single-cell transcriptomics has become a valuable approach for investigating IRD pathogenesis. By utilising its cellular-level resolution, several studies have elucidated how individual retinal cell populations respond at various stages of the degenerative process. Across diverse IRDs, scRNA-seq consistently revealed cell-type-specific disease mechanisms and heterogeneity (Figure 2), demonstrating how multiple retinal cell types respond differently at various stages of degeneration.

Single-cell approaches confirmed photoreceptor degeneration with deficits in phototransduction genes across models. Misfolded protein accumulation, heightened ER stress, and metabolic dysfunction emerged in degenerating photoreceptors across multiple models, leading to cell-intrinsic stress responses. These cellular stress responses triggered inflammatory cascades demonstrated across multiple studies, often involving microglia activation and Müller glia reactivity. Studies consistently revealed excessive immune-related signalling and inflammation driven by glial cells. Multiple studies demonstrated the dual capacity of Müller glia to shift between neuroprotective and proinflammatory roles. Understanding glial activity is therefore crucial for therapeutic approaches.

Multiple studies combined scRNA-seq with treatments and showed rescue of photoreceptor gene expression, improved survival, and reduced markers of stress or inflammation. Several research groups highlight the importance of early interventions for better outcomes. Single-cell analyses have identified novel therapeutic targets, including specific pathways and cell-type-specific vulnerabilities that could guide precision medicine approaches.

## 5. Challenges and Future Considerations

The field has benefited from rapid advances in single-cell platforms, yet the extensive genetic heterogeneity underlying IRDs, with hundreds of causative mutations, combined with variable disease onset and progression, creates substantial analytical complexity.

Retinal tissue processing presents unique challenges due to the elaborate morphology of photoreceptors and other retinal cells, making them particularly vulnerable during dissociation protocols. Given the extensive use of mouse models for single-cell transcriptomics in IRD research (12 out of 23 studies), small tissue samples such as mouse retinas present further complications due to low RNA yields, which can introduce sampling bias and reduce the reliability of gene expression measurements [160,161]. A technical optimisation is to increase the number of polymerase chain reaction (PCR) cycles during complementary DNA (cDNA) amplification, although this might increase PCR artifacts [161]. To address the challenges of low-input samples, several alternative scRNA-seq platforms have been developed that are especially well-suited for low-yield samples [162], such as the deterministic, mRNA-capture bead and cell co-encapsulation dropleting system, DisCo [163], and the open-source hydrogel droplet platform, HyDrop [164]. Additionally, increasing the number of biological replicates enables validation of cell clusters based on their reproducibility across samples, thereby preventing over-interpretation of clusters that may arise from technical variation [165]. To address sample preservation challenges, methodological innovations such as methanol fixation protocols have been developed to facilitate sample transport and storage whilst maintaining RNA integrity [166,167]. Single-nucleus RNA-seq represents an alternative approach that can be applied to frozen tissues [168]. Established protocols [166,167] are essential for minimising cell death and preserving cellular integrity. Batch effects can obscure genuine biological differences, and whilst computational tools can partially address technical variation [12], multiple biological replicates are essential for a robust approach, though this is often constrained by the high cost of single-cell experiments. Statistical interpretation requires careful consideration, as the large number of cells per condition can artificially inflate statistical power, potentially detecting differences that may not represent reproducible biological variation, particularly in studies with limited biological replicates [169,170,171]. Cell abundance interpretations require particular caution, as scRNA-seq-derived cell numbers may not accurately reflect the original tissue composition due to processing biases during tissue dissociation, technical variability, and quality control filtering steps [168] and should be validated through complementary methods such as flow cytometry or imaging.

Analysing scRNA-seq data presents challenges due to sparse gene expression, technical noise, and the complexity of heterogeneous cell populations [12]. Fortunately, sophisticated computational frameworks address these complexities, including the widely used Seurat pipeline [172,173] and numerous specialised packages within the Bioconductor project [174,175]. This open-source ecosystem allows researchers to inspect and modify analytical code, promoting transparency and reproducibility in single-cell research.

## 6. Conclusions

Although the application of single-cell transcriptomics remains relatively limited in IRD research, the field is rapidly evolving and showing considerable promise. The IRD studies presented in this review revealed the cellular complexity of the retinal tissue and demonstrated how single-cell approaches enabled the detection of early molecular changes preceding cell death. Across multiple IRD studies, researchers have identified common pathomechanisms, including disrupted phototransduction cascades, altered metabolic pathways, and compromised cellular stress responses across retinal cell types, with Müller glia consistently playing a crucial role during retinal degeneration.

These findings demonstrate how single-cell transcriptomics is advancing both mechanistic understanding and therapeutic development for IRDs. Future studies should consider combining single-cell and bulk RNA-seq to leverage the complementary strengths of both techniques, as bulk RNA-seq generally provides a higher sequencing depth per sample, while single-cell approaches reveal cellular heterogeneity [176,177]. Innovative analytical approaches, such as the deconvolution method applied by Lyu et al. [178] to bulk RNA-seq data using scRNA-seq references, demonstrate how these techniques can be combined to estimate cell-type proportions within bulk profiles.

Emerging advances in spatial multi-omics approaches promise to preserve tissue architecture whilst providing single-cell resolution, enabling researchers to understand not only gene expression in individual cells, but also how spatial organisation and cell–cell interactions contribute to disease progression [179,180,181]. Computational methods are now being developed to integrate scRNA-seq data with spatial transcriptomics data, offering powerful approaches to understand dynamic changes in complex tissue architecture during disease progression [182,183]. These integrated analytical frameworks are particularly valuable for IRDs, where understanding the spatial and temporal context of cellular dysfunction, such as the progression of photoreceptor degeneration from specific retinal regions or the spatial patterns of glial activation over time, provides crucial insights into disease mechanisms and optimal intervention points [184,185]. Moreover, spatial transcriptomics shows particular promise for evaluating therapeutic interventions, as it can capture regional variations in treatment efficacy and identify areas of the retina that respond differentially to gene therapy or pharmacological treatments over time [184,186]. These insights can inform optimisation of delivery and dosing strategies, ultimately making therapies more precisely targeted and potentially more effective. As scRNA-seq becomes increasingly accessible, its integration with established genetic and clinical knowledge will enhance our understanding of disease processes and enable more precise therapeutic interventions. The field would benefit from broader adoption of single-cell approaches in IRD research, supported by collaborative efforts to improve data sharing that enable large-scale integrative analyses.

## Figures and Tables

**Figure 1 genes-16-01088-f001:**
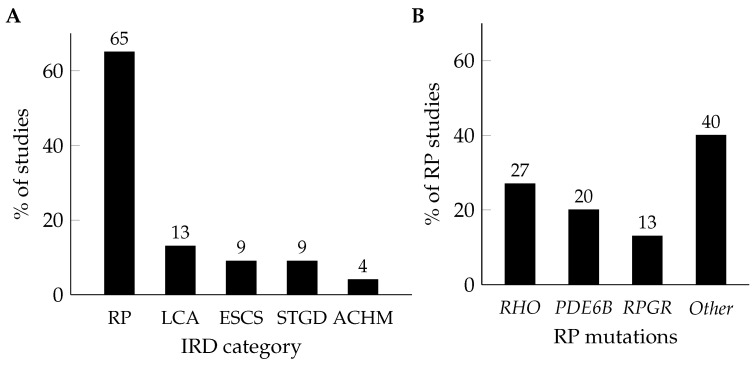
Distribution of single-cell transcriptomics studies in IRDs: (**A**) Overall distribution of studies across five major IRD categories: retinitis pigmentosa (RP; n = 15; 65%), Leber congenital amaurosis (LCA; n = 3; 13%), enhanced S-cone syndrome (ESCS; n = 2; 9%), Stargardt disease (STGD; n = 2; 9%), and achromatopsia (ACHM; n = 1; 4%). (**B**) Breakdown of RP studies by specific genetic mutations: *RHO* (n = 4; 27%), *PDE6B* (n = 3; 20%), *RPGR* (n = 2; 13%), and other RP-related mutations (n = 6; 40%). Data from literature search of 23 studies. ACHM: Achromatopsia; ESCS: Enhanced S-cone syndrome; IRD: Inherited retinal dystrophy; LCA: Leber congenital amaurosis; RP: Retinitis pigmentosa; STGD: Stargardt disease.

**Figure 2 genes-16-01088-f002:**
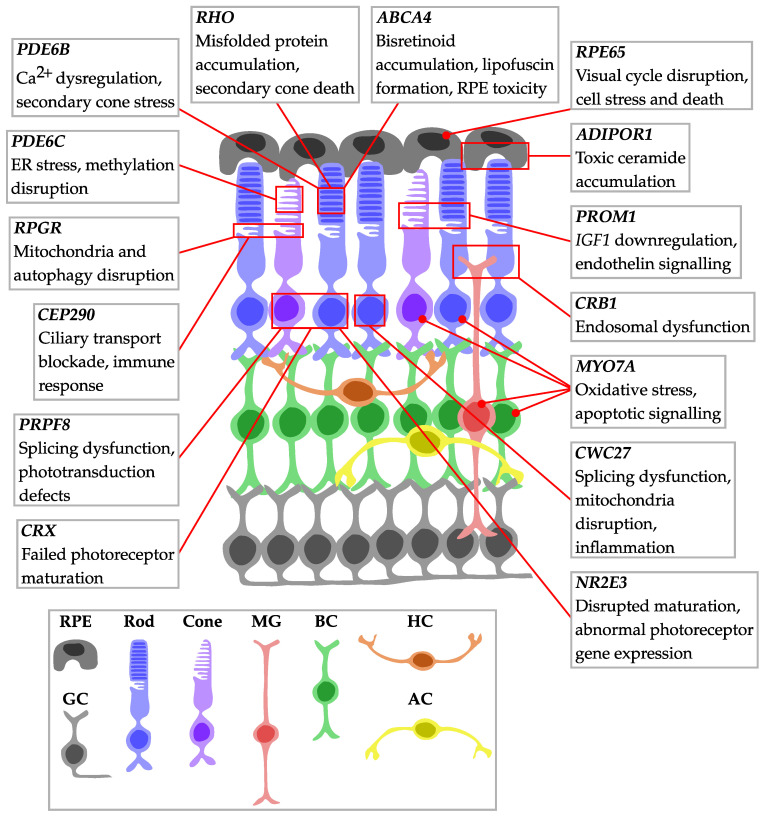
Schematic of the retina showing the locations of IRD-associated genes and key pathway dysregulation identified by scRNA-seq studies. Gene positions indicate primary expression sites, with disease mechanisms and findings from single-cell approaches including photoreceptor-specific vulnerabilities, glial responses, and therapeutic targets. Note that *MYO7A* localisation reflects scRNA-seq expression data, which differs from previous studies localising it primarily to the RPE. AC: Amacrine cell; BC: Bipolar cell; GC: Ganglion cell; HC: Horizontal cell; MG: Müller glia; RPE: Retinal pigment epithelium.

**Table 1 genes-16-01088-t001:** Single-cell transcriptomics studies examining pathological changes in disease models of inherited retinal dystrophies.

IRD	Mutation	Study	Model	Biological Replicates	Disease Stage	Intervention	Platform	Main Dysregulated Pathways or Markers in Disease (or Treatment) Conditions
RP	*RHO*	Tomita et al. [16]	Mouse (*Rho^P23H^*)	4 MUT and 4 WT	Active degeneration (PW7)	No	inDrop	↓ in PRs: phototransduction, cilium, development, ATP processes, glycolysis; ↑ in Müller cells: PR maintenance, mitochondrial localisation and transport
RP	*RHO*	Fu et al. [17]	Mouse (*Rho^P23H^*)	3 FGF21-treated and 3 vehicle-treated CTRL	Active degeneration (PW10)	FGF21 treatment	inDrop	FGF21-treated Müller cells and astrocytes: axon development↑, synapse formation↑, SRF signalling↑
RP	*RHO*	Santhanam et al. [18]	Zebrafish	1 MUT and 1 WT for V2 and V3 each; 3 fish pooled each	Active degeneration (age 6–10 months)	No	10x	Oxidative stress, metabolic reprogramming, misfolded proteins, circadian rhythm disruption, synaptic remodelling, regenerative signalling
RP	*RHO*	Lin et al. [19]	Human iPSC-derived rOrgs	1 patient and 1 healthy CTRL at D120 and D270 each; 5–6 rOrgs each	Immature (D120) and mature (D270) stage	No	10x	Visual perception, PR development, phototransduction, rod maturation, immune and inflammatory signalling
RP	*PDE6B*	Chen et al. [20] ^1^	Mouse (*rd1*)	MUT and WT at P11, P13, P17; 3 animals per time point	Early (P11), peak rod degeneration (P13), late rod degeneration (P17)	No	10x	Visual perception, phototransduction, apoptosis, Ca^2+^ signalling, MAPK pathway, metabolic pathways
RP	*PDE6B*	Dong et al. [21] ^1^	Mouse (*rd1*)	MUT and WT at P11, P13, P17; 3 animals per time point	Early (P11), peak rod degeneration (P13), late rod degeneration (P17)	No	10x	*Egr1*↑ in PRs
RP	*PDE6B*	Dong et al. [22] ^1^	Mouse (*rd1*)	MUT and WT at P11, P13, P17; 3 animals per time point	Early (P11), peak rod degeneration (P13), late rod degeneration (P17)	No	10x	*Hdac1*↑ in rods, *Parp1*↓ in cones
RP	*PDE6B*	Yan et al. [23] ^1^	Mouse (*rd1*)	MUT and WT at P11, P13, P17; 3 animals per time point	Focus on peak rod degeneration (P13)	No	10x	Ca^2+^ signalling in PRs
RP	*PDE6B*	Liao et al. [24]	Mouse (*rd1*)	2 MUT and 2 WT	Peak rod degeneration (P15)	No	10x	JNK signalling↑, *Jun* transcription factor↑
RP	*PDE6B*	Karademir et al. [25] ^2^	Mouse (*rd10*)	2 MUT and 2 WT	Peak rod degeneration (P21)	No	10x	Early phase: Ca^2+^ signalling, metabolic disruption, phototransduction, *Egr1* activation; late phase: mitochondrial respiratory dysfunction, synaptic remodelling, structural changes, *Cebpd* activation
RP	*PDE6B*	Sigurdsson and Grimm [26] ^2^	Mouse (*rd10*)	2 MUT and 2 WT	Peak rod degeneration (P21)	No	10x	Müller cell response: gliosis and metabolic markers, immune response, MHC components
RP, CRD, CD, MD	*RPGR*	Newton et al. [27]	Mouse (*Rpgr^ORFd5^*, *Rpgr^Ex3d8^*)	1 MUT and 1 WT for *Rpgr^ORFd5^* and *Rpgr^Ex3d8^* each	Active photoreceptor degeneration (18 months)	No	10x	↓ in PRs: phototransduction; ↑ in PRs: PI3K/AKT pathway, autophagy, necroptosis, mitochondrial function, TNF-α/NF-κB signalling, lysosome biogenesis
RP	*RPGR*	Li et al. [28]	Human iPSC-derived rOrgs	1 patient and 1 healthy CTRL at D40, D90, D150, D200 each; 2–3 rOrgs each	Multiple developmental time points (D40, D90, D150, D200)	No	10x	↑ of PR markers
RP	*MYO7A*	Leong et al. [29]	Human iPSC-derived rOrgs	3 patients and 3 healthy CTRLs	Early/pre-degenerative stage (D245/35 weeks)	No	10x	↑ in rods: cellular and oxidative stress, hydrogen peroxide metabolism; no pathology detected for cones; ↑ in Müller cells: apoptosis
RP	*CWC27*	Bertrand et al. [30]	Mouse (*Cwc27^K338fs^*)	1 MUT and 1 WT	Early degenerative stage (4 months)	No	10x	↓ in rods: mitochondrial genes; ↑ in Müller cells: inflammation-related genes
RP	*PRPF8*	Atkinson et al. [31]	Human iPSC-derived rOrgs	4 patients and 4 isogenic CTRLs	Advanced degeneration (D210)	No	10x	*MALAT1*↑ (degenerating rods), ↓ of cone (*ARR3*) and rod (*NRL*) markers, ↓ of phototransduction and mitochondrial genes, ciliary dysfunction
RP, LCA	*CRB1*	Boon et al. [32] ^3^	Human iPSC-derived rOrgs	1 patient; 1^st^ scRNA-seq: 1 patient and 1 healthy CTRL; 2^nd^ scRNA-seq: 1 gene therapy-treated (gene 1) and 1 gene therapy-treated (gene 2) and 1 CTRL-treated; 5–6 rOrgs each	Active photoreceptor degeneration (D230)	AAV gene therapy	10x	Untreated disease: endosomal system dysfunction in rods and Müller cells; phototransduction cascade activation in rods; cell adhesion, protein binding, cell death, iron ion transport in Müller cells; AAV-treated: partial restoration of endosomal system pathways toward control levels
RP, MD	*PROM1*	Shigesada et al. [33]	Mouse (Prom1^−/−^)	1 light-exposed MUT and 1 dark-reared MUT	Early/pre-symptomatic stage (P11)	No	10x	*Igf1*↓ in rods and astrocytes, *Edn2*↑ in rods, glial activation in Müller glia and astrocytes, visual function dysfunction in PRs
RP	*ADIPOR1*	Lewandowski et al. [34]	Mouse (AdiporR1^−/−^)	2 MUT and 2 WT at P19, P25, P30 each	Early/pre-onset (P18), active/post-onset (P25), advanced (P30) degeneration	No	10x	Pre-onset: translation↑, oxidative phosphorylation↑, mitochondrion organisation↑, RNA splicing↓, protein localisation↓; active and advanced phase: visual perception↓, translation↑; neurodegenerative pathways
LCA	*CEP290*	Fogerty et al. [35]	Zebrafish	5 MUT and 5 HET CTRLs	Active degeneration (6 months)	No	SplitBio	↑ in Müller cells: *notch3*, *stat2*, *yap1*, rest-associated genes
LCA	*RPE65*	Choi et al. [36]	Mouse (*rd12*)	4 treated MUT and 4 untreated MUT and 4 WT	Advanced degeneration (2 months)	Base editing therapy	10x	Base editing-treated cones: phototransduction restoration (rescue of phototransduction genes), opsin recovery (*Opn1sw*), neuroprotection (*Mt1*↑), cell survival (cell death/stress genes↓)
LCA, CRD, MD	*CRX*	Kruczek et al. [37]	Human iPSC-derived rOrgs	2 patients (treated and untreated) and 2 healthy CTRLs; 3–4 rOrgs each	Impaired photoreceptor maturation (D200)	AAV gene therapy	10x	AAV-CRX-treated PRs: partial rescue of *RHO*, *OPN1MW*, *CABP4*
ESCS	*NR2E3*	Mullin et al. [38]	Human iPSC-derived rOrgs	1 patient and 1 isogenic CTRL and 1 healthy CTRL at D40, D80, D120, D160, D260 each; 10 rOrgs each	Multiple time points (D40, D80, D120, D160, D260)	AAV gene therapy	10x	Untreated PRs: phototransduction pathway disruption; “divergent rods”: *RHO*↓, cone genes↑ (*ARR3*, *PDE6H*, *GNAT2*), rod genes↑ (*GNAT1*)
ESCS	*NR2E3*	Aísa-Marín et al. [39]	Mouse (*Nr2e3* Δ27, *Nr2e3* ΔE8)	P40: 2 ΔE8 MUT and 2 WT; P80: 2 Δ27 MUT and 1 WT	Early-onset, but stable retinal dysfunction (Δ27 at P80); pre-symptomatic stage of progressive late-onset retinal degeneration (ΔE8 at P40)	No	10x	Formation of “hybrid photoreceptors”, phototransduction↑, autophagy and necrosis↑, stress response↑, homeostasis↓, mitochondria and synaptic function↓
STGD	*ABCA4*	Watson et al. [40]	Human iPSC-derived rOrgs	3 patients and 2 healthy CTRLs; at least 25 rOrgs each	Impaired photoreceptor maturation (D200)	No	10x	Stress response pathways (mTOR signalling, mitochondrial dysfunction, oxidative phosphorylation), phototransduction disruption, cell cycle dysregulation, and impaired maturation
STGD	*ABCA4*	Zhao et al. [41]	Human iPSC-derived rOrgs	1 patient and 1 healthy CTRL	Multiple time points (D40, D90, D150, D200, D260)	No	10x	No specific pathways or markers reported
ACHM	*PDE6C*	Miller et al. [42]	Mouse (*Pde6c^cpfl1^*)	1 treated MUT and 1 untreated MUT	Peak cone degeneration (P24)	H3K27 demethylase inhibitor treatment	SMART-seq2 / MARS-seq	GSK-J4-treated cones: oxidative phosphorylation↓, mitochondrial dysfunction↓, endoplasmic reticulum stress↓

^1^ Studies used same scRNA-seq dataset. GSE ID: GSE212183. ^2^ Studies used same scRNA-seq dataset. GSE ID: GSE183206. ^3^ Authors performed two separate scRNA-seq experiments on P128 patient samples as well as analyses on organoids from P116 and P117 patients to support their findings from P128 samples (not included in this table). Abbreviations: ↓: Downregulated; ↑: Upregulated; AAV: Adeno-associated virus; CD: Cone dystrophy; CRD: Cone-rod dystrophy; CTRL: Control; D: Day; EGR1: Early growth response 1; ESCS: Enhanced S-cone syndrome; FGF21: Fibroblast growth factor 21; GSE ID: Gene expression omnibus series identifier; H3K27: Histone H3 lysine 27; HDAC: Histone deacetylase; HET: Heterozygous; iPSCs: Induced pluripotent stem cells; LCA: Leber congenital amaurosis; MAPK: Mitogen-activated protein kinase; MD: Macular dystrophy; MHC: Major histocompatibility complex; MUT: Mutant; P: Postnatal day; PARP: Poly (ADP-ribose) polymerase; PR: Photoreceptor(s); PW: Postnatal week; rOrgs: Retinal organoids; RP: Retinitis pigmentosa; scRNA-seq: Single-cell RNA sequencing; SRF: Serum response factor; V: Version; WT: Wild-type.

## Data Availability

The original contributions presented in this study are included in the article. Further inquiries can be directed to the corresponding authors.

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
