# Peer review of "Single-Cell Transcriptomics in Inherited Retinal Dystrophies: Current Findings and Emerging Perspectives"

_genes, 2025, doi:10.3390/genes16091088_

Round 1
Reviewer 1 Report
Comments and Suggestions for Authors
In this review, Nguyen et al. discuss the use of scRNA-seq to study inherited retinal dystrophies (IRDs). Unlike the conventional methods, analyzing individual cells using this approach is crucial for understanding how the retinal cells behave during the disease progression. The authors discusses genetic signatures, molecular changes, and the disease mechanisms (like photoreceptor degeneration and inflammation) cross the IRD subtypes. The review also discuss how scRNA-seq can effectively applied and combine with other techniques for advancing our understanding of IRD pathogenesis and developing new therapies.
The review is well-written, covers all relevant aspects of IRD, and requires no revisions. I've only spotted a minor repetition of the word "reduced" [line 117]
Reviewer 2 Report
Comments and Suggestions for Authors
Thank you for the opportunity to review this manuscript entitled, "Single-cell transcriptomics in inherited retinal dystrophies: Current findings and emerging perspectives". In this comprehensive review, the authors systematically survey the recent literature on the application of single-cell transcriptomic techniques to the study of Inherited Retinal Dystrophies (IRDs). The authors summarize key findings across a range of IRDs, identify common pathomechanisms, and discuss the therapeutic implications and future challenges of this powerful technology.
This is an exceptionally well-researched, well-organized, and timely review that provides a significant contribution to the field. The pace of discovery in single-cell genomics is rapid, and a synthesis of its application to IRDs is much needed. The strengths of this manuscript are numerous:
- The literature search is systematic and clearly described.
- The organization of the findings by disease and gene is logical and highly effective for the reader.
- Table 1 is an outstanding resource that meticulously summarizes the key features of the 23 included studies, which will be of great value to researchers in this area.
- Figure 2 provides an excellent and intuitive visual summary of the key gene- and cell-type-specific pathway dysregulations identified across the studies reviewed.
I have no major concerns regarding the manuscript and believe it should be published as it will serve as an essential reference for clinicians and scientists working to unravel the complexities of IRDs and develop novel therapies. I am grateful for the opportunity to review this work.
Reviewer 3 Report
Comments and Suggestions for Authors
The review manuscript on Single cell transcriptomics in Inherited retinal dystrophies is a great piece of work. It is a well-written, comprehensive review with up-to-date citations. Authors have done a great job compiling this information on single-cell studies. This review summarizes the single-cell transcriptomic findings reported on different animal models of genetic retinal dystrophies and organoids along with the studied age-points, that would be very relevant for the fellow researchers.
I only have a few things to comment upon:
- Table 1: Addition of scRNA-seq method utilized by each of the mentioned study as a separate column will be very helpful. I understand there is not enough space, but I would suggest incorporation of another column for ‘markers or pathways identified/reported’ by each study, if feasible.
- Challenges and future considerations (section 5); line 588-589: Authors must add few technical challenges faced with small tissues like mice retina, such as low starting RNA amount that could introduce bias, considering mice models are extensively used.
- Section 6; line 629-632: Nowadays, methods to integrate the sc-seq data and spatial data are being developed which could help understand complex tissue architecture. A few lines highlighting their importance could be added here.
